# Palmitic Acid-Conjugated Radiopharmaceutical for Integrin α_v_β_3_-Targeted Radionuclide Therapy

**DOI:** 10.3390/pharmaceutics14071327

**Published:** 2022-06-23

**Authors:** Guangjie Yang, Hannan Gao, Chuangwei Luo, Xiaoyu Zhao, Qi Luo, Jiyun Shi, Fan Wang

**Affiliations:** 1Medical Isotopes Research Center and Department of Radiation Medicine, State Key Laboratory of Natural and Biomimetic Drugs, School of Basic Medical Sciences, Peking University, Beijing 100191, China; docyang@bjmu.edu.cn (G.Y.); luocw@bjmu.edu.cn (C.L.); 2011110025@bjmu.edu.cn (X.Z.); 2Key Laboratory of Protein and Peptide Pharmaceuticals, CAS Center for Excellence in Biomacromolecules, Institute of Biophysics, Chinese Academy of Sciences, Beijing 100101, China; gaohannan@ibp.ac.cn; 3Guangzhou Laboratory, Guangzhou 510005, China; luo_qi@gzlab.ac.cn

**Keywords:** ^177^Lu, RGD (Arg-Gly-Asp), albumin binder, palmitic acid, peptide receptor radionuclide therapy (PRRT), tumor

## Abstract

Peptide receptor radionuclide therapy (PRRT) is an emerging approach for patients with unresectable or metastatic tumors. Our previously optimized RGD peptide (3PRGD_2_) has excellent targeting specificity for a variety of integrin α_v_β_3_/α_v_β_5_-positive tumors and has been labeled with the therapeutic radionuclide [^177^Lu]LuCl_3_ for targeted radiotherapy of tumors. However, the rapid clearance of [^177^Lu]Lu-DOTA-3PRGD_2_ (^177^Lu-3PRGD_2_) in vivo requires two doses of 111 MBq/3 mCi to achieve effective tumor suppression, limiting its further clinical application. Albumin binders have been attached to drugs to facilitate binding to albumin in vivo to prolong the drug half-life in plasma and obtain long-term effects. In this study, we modified 3PRGD_2_ with albumin-binding palmitic acid (Palm-3PRGD_2_) and then radiolabeled Palm-3PRGD_2_ with ^177^Lu. [^177^Lu]Lu-DOTA-Palm-3PRGD_2_ (^177^Lu-Palm-3PRGD_2_) retained a specific binding affinity for integrin α_v_β_3_/α_v_β_5_, with an IC_50_ value of 5.13 ± 1.16 nM. Compared with ^177^Lu-3PRGD_2_, the ^177^Lu-Palm-3PRGD_2_ circulation time in blood was more than 6 times longer (slow half-life: 73.42 min versus 11.81 min), and the tumor uptake increased more than fivefold (21.34 ± 4.65 %IA/g and 4.11 ± 0.70 %IA/g at 12 h post-injection). Thus, the significant increase in tumor uptake and tumor retention resulted in enhanced efficacy of targeted radiotherapy, and tumor growth was completely inhibited by a single and relatively lowdose of 18.5 MBq/0.5 mCi. Thus, ^177^Lu-Palm-3PRGD_2_ shows great potential for clinical application.

## 1. Introduction

Peptide receptor radionuclide therapy (PRRT) is a promising treatment for patients with unresectable or metastatic tumors and has been widely used in clinical practice [1]. Currently, somatostatin analog (SSA)-mediated radionuclide targeting therapy is the most commonly used PRRT in clinical practice and has been approved for clinical application in some countries. Although some progress has made in PRRT applied in clinical practice, most radionuclide-labeled peptides for targeted radiotherapy still face insufficient tumor uptake, and the retention of labeled peptides in tumors is not sufficient, thus limiting their therapeutic effect.

Albumin makes up approximately 55–60% of serum proteins, and its biological half-life is reportedly 19 days, which makes it possible to leverage the long circulation half-life of albumin to produce long-acting therapeutics [2,3,4]. Radiopharmaceuticals conjugated with albumin binders, such as 4-(p-iodophenyl)butyric acid and truncated Evans Blue (EB), have been used in radiotherapy targeting folate receptor [5,6,7], somatostatin receptor (SSTR) [8,9], integrin α_v_β_3_/α_v_β_5_ [10] and prostate-specific membrane antigen (PSMA) [11,12,13,14] and have achieved significantly prolonged blood circulation, with enhanced tumor suppression [3,15]. In addition to these well-studied albumin binding groups, fatty acids have also been demonstrated to be typical albumin binding groups and are easy to modify, have affinity of various strengths with albumin and have strong cell membrane penetration ability [16,17,18,19]. Recently, Zhang et al., successfully improved the tumor uptake and retention of fibroblast activation protein inhibitor (FAPI) tracers by introducing fatty acids (lauric acid and palmitic acid), as albumin binders [20]. Fatty acid-conjugated peptide drugs, such as Levemir, Tresiba and liraglutide, have been approved by the FDA for clinical use, and they have been found to achieve long-term effects through the binding of fatty acids to albumin, prolonging the blood circulation of insulin and glucagon-like peptide-1 analogs [21]. Therefore, the use of fatty acids as albumin binders is a promising strategy for development of peptide-based radiopharmaceuticals as long-term tumor-targeted radiotherapy agents for clinical application.

Integrin α_v_β_3_/α_v_β_5_ is highly upregulated during tumor angiogenesis and in some tumor cells but not in quiescent vessels and normal organ systems, making it an ideal tumor target for receptor-mediated broad-spectrum tumor-targeting imaging and therapy [22,23]. Many integrin α_v_β_3_/α_v_β_5_-targeted RGD peptide radiopharmaceuticals have been developed and are widely used in clinical practice [24,25,26,27]. Our ^99m^Tc-labeled 3PRGD_2_ has been clinically used as a diagnostic tracer for early detection of various tumors, and its clinical phase III study has been completed [28,29,30]. In a previous study, we prepared ^177^Lu-labeled 3PRGD_2_ and carried out PRRT studies in animal model. Although as a targeted radiotherapy agent, ^177^Lu-labeled 3PRGD_2_ showed considerable therapeutic efficacy in mouse tumor model, it must be administered twice at a dose of 111 MBq or in combination with Endostar chemotherapy [31]. Its short plasma half-life as well as insufficient tumor uptake and retention time limit its further clinical application. Therefore, we wondered whether introduction of fatty acids could improve the therapeutic effect of ^177^Lu-3PRGD_2_.

Since palmitic acid is the most common fatty acid used for drug modification to obtain long circulation drugs, we introduced palmitic acid (termed palm) into the [^177^Lu]Lu-DOTA-3PRGD_2_ structure to obtain [^177^Lu]Lu-DOTA-Palm-3PRGD_2_ and evaluated its potential for targeted radiotherapy in a mouse tumor model. We expected that the introduction of palmitic acid would significantly prolong the blood half-life of ^177^Lu-3PRGD_2_ and significantly increase the effective dose of ^177^Lu-3PRGD_2_ and its duration of action in tumors, thus achieving complete tumor elimination at a low dose. Due to the broad-spectrum expression of the integrin receptor, this long-acting RGD radiotherapy agent has broad application prospects in targeted radiotherapy for a variety of tumors.

## 2. Materials and Methods

### 2.1. Materials

Chemicals and solvents were purchased from Sigma-Aldrich (St. Louis, MO, USA). Fmoc-Lys(palmitoyl-Glu-OtBu)-OH was purchased from GlpBio (Montclair, CA, USA). The bifunctional chelator 1,4,7,10-tetraazacyclododecane-1,4,7,10-tetraacetic acid mono-N-hydroxysuccinimide ester (DOTA-NHS ester) was purchased from Macrocyclics Inc. (Dallas, TX, USA). PEG_4_-E[PEG_4_-c(RGDfK)]_2_ (termed 3PRGD_2_) was obtained from CS BIO (Menlo Park, CA, USA). ^177^LuCl_3_ solution was purchased from ITG (Schwaig, Germany).

### 2.2. Chemical Synthesis of Conjugates

The synthetic routes of DOTA-Lys(palmitoyl-Glu-OH)-3PRGD_2_ (termed DOTA-Palm-3PRGD_2_), DOTA-3PRGD_2_, and DOTA-Lys(palmitoyl-Glu-OH)-OH (termed DOTA-Palm) are shown in the Appendix A. The preparation procedures are described below. The products were analyzed and isolated using an Agilent 1260 HPLC system equipped with a semipreparative C4 column (Sepax Bio-C4, 10 mm × 250 mm, 5 μm) and a UV/Vis detector (λ = 210 nm or 254 nm). The mobile phase was composed of phase A (0.05% TFA in water) and phase B (0.05% TFA in acetonitrile). The flow rate was 3.2 mL/min, and the phase B gradients are described in the preparation procedures.

#### 2.2.1. Synthesis of DOTA-Palm-3PRGD_2_

Synthesis of Fmoc-Lys(palmitoyl-Glu-OtBu)-NHS. Fmoc-Lys(palmitoyl-Glu-OtBu)-OH (20.0 mg, 1.0 eq), 1-ethyl-3-(3-(dimethylamino)propyl)-carbodiimide hydrochloride (EDC·HCl) (7.3 mg, 1.5 eq), and N-hydroxysuccinimide (5.8 mg, 2.0 eq) were dissolved in 1.0 mL dimethylformamide (DMF). The reaction mixture was stirred at room temperature overnight. The product was separated via HPLC (the phase B gradient started from 45% at 0 min to 80% at 25 min and was increased to 45% at 30 min, Method 1). The fraction at 25.8 min was collected and lyophilized to afford Fmoc-Lys(palmitoyl-Glu-OtBu)-NHS. The yield was 19.3 mg (~86%). ESI-MS: *m*/*z* = 912.72 for [M + Na]^+^ (M = 889.14 calcd for [C_50_H_72_N_4_O_10_]).

Synthesis of Lys(palmitoyl-Glu-OH)-3PRGD_2_. Fmoc-Lys(palmitoyl-Glu-OtBu)-NHS (5.0 mg, 1.0 eq) and 3PRGD_2_ (11.6 mg, 1 eq) were dissolved in 200 μL DMF. After the addition of DIEA to adjust the solution pH to 8.0, the mixture was stirred at room temperature overnight. The product was separated via HPLC (Method 1), and the fraction at 17.6 min was collected and lyophilized to afford Fmoc-Lys(palmitoyl-Glu-OtBu)-3PRGD_2_. The product (5 mg) was dissolved in 400 μL of TFA, stirred at room temperature for 5 min and then blown dry with nitrogen. The reaction product was dissolved in 100 μL DMF. After addition of 25 μL piperidine, the mixture was stirred at room temperature for 10 min. The product was separated via HPLC (Method 1). The fraction at 12.7 min was collected and lyophilized to afford Lys(palmitoyl-Glu-OH)-3PRGD_2_. The yield was 2.7 mg (~60%). MALDI-TOF-MS: *m*/*z* = 2555.71 for [M + H]^+^ (M = 2554.45 calcd for [C_119_H_199_N_25_O_36_]).

Synthesis of DOTA-Palm-3PRGD_2_. Lys(palmitoyl-Glu-OH)-3PRGD_2_ (1.3 mg, 1 eq) and DOTA-NHS ester (0.41 mg, 1.6 eq) were dissolved in 200 μL DMF. After the addition of DIEA to adjust the solution pH to 8.0, the mixture was stirred at room temperature overnight. The product was separated via HPLC (Method 1), and the fraction at 11.9 min was collected and lyophilized to afford DOTA-Palm-3PRGD_2_. The yield was 0.6 mg (~40%). MALDI-TOF-MS: *m*/*z* = 2941.25 for [M + H]^+^ (M = 2940.63 calcd for [C_135_H_225_N_29_O_43_]).

#### 2.2.2. Synthesis of DOTA-3PRGD_2_

3PRGD_2_ (2.0 mg, 1 eq) and DOTA-NHS ester (0.73 mg, 1.5 eq) were dissolved in 200 μL DMF. After the addition of DIEA to adjust the solution pH to 8.0, the mixture was stirred at room temperature overnight. The product was separated via HPLC. The mobile phase was isocratic with 90% phase A and 10% phase B at 0–5 min, followed by a mobile phase gradient from 10% phase B at 5 min to 60% at 25 min and to 10% at 30 min (Method 2). The fraction at 18.0 min was collected and lyophilized to afford DOTA-3PRGD_2_. The yield was 1.2 mg (~50%). MALDI-TOF-MS: *m*/*z* = 2446.02 for [M + H]^+^ (M = 2445.26 calcd for [C_108_H_176_N_26_O_38_]).

#### 2.2.3. Synthesis of DOTA-Palm

Fmoc-Lys(palmitoyl-Glu-OtBu)-OH (5 mg) was dissolved in 200 μL TFA. The mixture was stirred at room temperature for 5 min and then blown dry with nitrogen. The reaction product was dissolved in 100 μL DMF. After addition of 25 μL piperidine, the mixture was stirred at room temperature for 10 min. The product was separated by HPLC. The mobile phase was isocratic with 70% phase A and 30% phase B at 0–5 min, followed by a mobile phase gradient from 30% phase B at 5 min to 80% at 25 min and to 30% at 30 min (Method 3). The fraction at 19.5 min was collected and lyophilized to afford Lys(palmitoyl-Glu-OH)-OH. The product (1.0 mg, 1 eq) and DOTA-NHS ester (1.46 mg, 1.5 eq) were dissolved in 200 μL DMF. After addition of DIEA to adjust the solution pH to 8.0, the mixture was stirred at room temperature overnight. The product was separated via HPLC (Method 3). The fraction at 18.9 min was collected and lyophilized to afford DOTA-Palm. The yield was 0.8 mg (~45.7%). MALDI-TOF-MS: *m*/*z* = 900.41 for [M]^+^ (M = 899.56 calcd for [C_43_H_77_N_7_O_13_]).

### 2.3. Radiochemistry and In Vivo Stability

DOTA-Palm-3PRGD_2_ (40 μg/5 μL DMSO, 13.6 nmol), DOTA-3PRGD_2_ (40 μg/5 μL H_2_O, 16.3 nmol), or DOTA-Palm (15 μg/5 μL DMSO, 16.7 nmol) was added to a mixture of 200 μL NH_4_OAc buffer (0.1 M, pH = 4.8) and 20 μL [^177^Lu]LuCl_3_ solution (~185 MBq/5 mCi). Next, the vials were heated in an air bath at 100 °C for 25 min. After cooling to room temperature, the radiopharmaceuticals were analyzed with an Agilent 1260 HPLC system equipped with a radioactive detector and a C18 column (YMC-Pack ODS-A, 250 × 4.6 mml.D. S-5 μm, 12 nm). The flow rate was 1 mL/min. The gradient mobile phase started from 30% phase B at 0 min and progressed to 70% phase B at 25 min and 30% phase B at 30 min. To evaluate in vivo stability, ^177^Lu-Palm-3PRGD_2_ (37 MBq, 2.7 nmol) was injected via the tail vein, and urine samples were collected and analyzed using radio-HPLC at 1 and 6 h post-injection (p.i.).

### 2.4. n-Octanol/PBS Distribution Coefficient

The n-octanol/PBS distribution coefficients of ^177^Lu-Palm-3PRGD_2_, ^177^Lu-3PRGD_2_, and [^177^Lu]Lu-DOTA-Palm (^177^Lu-Palm) in an n-octanol/PBS system were determined as previously reported [32]. Briefly, the radiopharmaceuticals were prepared and purified with Sep-Pak-C18 cartridges and then dissolved in a mixed solution of 5 mL *n*-octanol and 5 mL PBS. After vortexing for 1 h, the mixture was centrifuged at 10,000 rpm for 10 min. Samples (100 μL, *n* = 4) from the n-octanol and PBS components were collected and counted using a γ-counter. The cpm values were calculated as the logarithm of the *n*-octanol/PBS ratio.

### 2.5. Blood Clearance Study

Female KM mice were randomly divided into three groups (*n* = 5). Each group was injected intravenously with 0.74 MBq ^177^Lu-Palm-3PRGD_2_, ^177^Lu-3PRGD_2_, or ^177^Lu-Palm. Blood samples obtained from the canthus vein were collected at different time points post injection (p.i.), weighed, and evaluated using a γ-counter. The results are presented as the percentage injected activity per gram (%IA/g). Nonlinear regression (curve fit) followed by two phase decay was performed using GraphPad Prism version 9.0.0 for Windows, GraphPad Software, San Diego, CA, USA, www.graphpad.com (accessed on 22 May 2022). And then the half-lives values were calculated and determined.

### 2.6. Cell Culture and Animal Model

The human glioma U87MG (ATCC^®^ HTB-14™) cell lines were purchased from the American Type Culture Collection (ATCC, Manassas, VA, USA). The murine colon adenocarcinoma MC38 cell lines were kindly provided by the lab of Prof. Yangxin Fu at the Institute of Biophysics, Chinese Academy of Sciences (Beijing, China). U87MG and MC38 cells were cultured in Dulbecco’s modified Eagle’s medium (DMEM) with 10% fetal bovine serum at 37 °C in a humidified atmosphere containing 5% CO_2_. Female C57BL/6 mice (6 weeks of age) were purchased from Beijing Vital River Laboratory Animal Technology Co., Ltd. An MC38 tumor model was established via subcutaneous injection of MC38 cells (1.0 × 10^6^) into the right front flank. When the tumor volume reached the size of 60~100 mm^3^, the mice were used for a targeted radionuclide therapy study. When the tumor volume reached 150~200 mm^3^, the mice were used for a biodistribution study and SPECT/CT imaging. All animal experiments were performed in accordance with the guidelines of the Institutional Animal Care and Use Committee (IACUC) of Peking University.

### 2.7. Cell Uptake Assay

Integrin α_v_β_3_/α_v_β_5_-positive U87MG glioma cells were seeded into a 6-well plate. Then, the cells were incubated with ^177^Lu-Palm-3PRGD_2_ (0.74 MBq, 0.054 nmol) or ^177^Lu-3PRGD_2_ (0.74 MBq, 0.065 nmol) in 1 mL fresh medium (containing the binding ions 20 mM Tris, 150 mM NaCl, 2 mM CaCl_2_, 1 mM MgCl_2_, and 1 mM MnCl_2_; pH = 7.4; without FBS) at 37 °C for 1, 4, or 24 h. Then, the medium was removed, and the cells were washed three times with ice-cold PBS. Finally, the cells were lysed with 0.5 mL 0.5 M NaOH twice, and the NaOH solution (0.5 mL × 2) was collected for γ-count determination. The results are presented as a percentage of the added dose per million cells (%AD/10^6^ cells).

### 2.8. Competition Binding Assays of ^177^Lu-Palm-3PRGD_2_ and ^177^Lu-3PRGD_2_

The binding affinity of DOTA-Palm-3PRGD_2_ and DOTA-3PRGD_2_ to integrin α_v_β_3_/α_v_β_5_ was determined using U87MG glioma cells. Filter multiscreen DV plates were seeded with 10^5^ U87MG cells in binding buffer (20 mM Tris, 150 mM NaCl, 2 mM CaCl_2_, 1 mM MgCl_2_, 1 mM MnCl_2_, pH = 7.4) and incubated at 4 °C with ^177^Lu-Palm-3PRGD_2_ or ^177^Lu-3PRGD_2_ in the presence of increasing concentrations of unlabeled 3PRGD_2_. Meanwhile, U87MG cells were also incubated with ^177^Lu-Palm-3PRGD_2_ and unlabeled 3PRGD_2_ in the presence of human serum albumin (HSA) at 4 °C. After removal of the unbound radiolabeled tracers and several washes with ice-cold PBS, the hydrophilic PVDF filters were collected. Radioactivity was determined using a γ-counter. Nonlinear regression (curve fit) followed by one-site specific binding was performed using GraphPad Prism version 9.0.0. Then the IC_50_ values were calculated and determined.

### 2.9. Biodistribution Study

Mice bearing MC38 xenografts were randomly divided into 10 groups (*n* = 4). The mice in five groups were administered 0.74 MBq/20 µCi of ^177^Lu-Palm-3PRGD_2_ and sacrificed at 1, 4, 12, 24, and 72 h p.i. The mice on one group was injected with 0.74 MBq/20 µCi of ^177^Lu-Palm-3PRGD_2_ and 500 μg (242 nmol) 3PRGD_2_ as a blocking agent and sacrificed at 1 h p.i. The mice in the four remaining groups were administered 0.74 MBq/20 µCi of ^177^Lu-Palm or ^177^Lu-3PRGD_2_ and sacrificed at 4 and 12 h p.i. Tumors and major organs were harvested, weighed and measured for radioactivity using a γ-counter. The organ uptake was calculated as the percentage injected activity per gram (%IA/g). The effective absorbed dose in humans estimated from mouse biodistribution data by using a dedicated software (OLINDA 1.0).

### 2.10. Small-Animal SPECT/CT Imaging

SPECT/CT imaging was performed using a small animal SPECT/CT imaging system (Mediso Inc., Budapest, Hungary). Each mouse bearing MC38 tumors was injected with a radiotracer at a radioactivity of 37 MBq/1 mCi. The mice were imaged at 1, 4, 12, 24, and 72 h after injection of ^177^Lu-Palm-3PRGD_2_ (37 MBq, 2.7 nmol), and the mice in the blocking study were imaged at 1 h p.i. The mice were imaged at 1 and 4 h after injection of ^177^Lu-3PRGD_2_ (37 MBq, 3.3 nmol) or ^177^Lu-Palm (37 MBq, 3.3 nmol). Pinhole SPECT images (peak, 56.1, 112.9, and 208.4 keV; 20% width; frame time, 25 s) were acquired, and CT images were subsequently acquired (50 kV; 0.67 mA; rotation, 210°; exposure time, 300 ms). The raw data were reconstructed in a whole-body region. The SPECT and CT images were then fused using Nucline v 2.01 (Mediso Inc., Budapest, Hungary). The maximum intensity projection (MIP) was determined for whole-body imaging from the posterior view.

### 2.11. Targeted Radionuclide Therapy

To assess and compare the therapeutic potential of ^177^Lu-Palm-3PRGD_2_, ^177^Lu-3PRGD_2_ and ^177^Lu-Palm, MC38 tumor models were used. MC38 tumor-bearing mice with a tumor size of 60~100 mm^3^ were randomly divided into four groups (6~8 mice/group). The mice were injected via the tail vein with a single dose injection of saline (as a control), 18.5 MBq/0.5 mCi of ^177^Lu-Palm-3PRGD_2_ (1.35 nmol), ^177^Lu-3PRGD_2_ (1.6 nmol) or ^177^Lu-Palm (1.7 nmol), respectively. Tumor dimensions and body weight were measured every two or three days. The tumor volume was calculated as 1/2(length × width × width). Mice were euthanized when the body weight lost >20% of the original weight. Major organs (heart, lung, liver, spleen and kidney) were harvested at the end of the treatment study and evaluated for potential toxicity using standard hematoxylin and eosin (H & E) staining analysis.

### 2.12. Statistical Analysis

Numerical results are reported as the mean ± standard deviation. Means were compared using Student’s *t*-test or multiple unpaired *t*-test. *p*-values < 0.05 were considered statistically significant. * indicates *p* < 0.05, ** indicates *p* < 0.01, *** indicates *p* < 0.001, and **** indicates *p* < 0.0001.

## 3. Results

### 3.1. Chemical Synthesis and Radiolabeling

Detailed synthetic results for DOTA-Palm-3PRGD_2_, DOTA-3PRGD_2_, and DOTA-Palm are shown in the Appendix A. DOTA-Palm-3PRGD_2_, DOTA-3PRGD_2_ and DOTA-Palm were obtained with more than 95% purity (Appendix A) and confirmed by MALDI-TOF-MS (Appendix A). All ^177^Lu-labeled radiopharmaceuticals were prepared by reacting ^177^LuCl_3_ with the respective DOTA conjugate (DOTA-Palm-3PRGD_2_, DOTA-3PRGD_2_, and DOTA-Palm) in NH_4_OAc buffer (0.1 M, pH = 4.8) at 100 °C for 25 min. Schematic structures of radiopharmaceuticals are shown in Figure 1. The radiochemical purities were >95%, with a molar activity of 13.6 MBq/0.37 mCi per nmol, 11.1 MBq/nmol and 11.3 MBq/nmol for ^177^Lu-Palm-3PRGD_2_, ^177^Lu-Palm and ^177^Lu-3PRGD_2_, respectively (Appendix A). ^177^Lu-Palm-3PRGD_2_ showed satisfactory in vivo stability (Appendix A).

### 3.2. n-Octanol/PBS Distribution Coefficient (logP_O/W_)

The log P_O/W_ values of ^177^Lu-Palm-3PRGD_2_, ^177^Lu-3PRGD_2_ and ^177^Lu-Palm were determined to be −1.25 ± 0.07, −4.05 ± 0.14 and 5.93 ± 0.01, respectively (Table 1). This result is consistent with the trend of the radiopharmaceutical retention time analyzed via radio-HPLC (22.07, 5.05 and 25.13 min, respectively). These results indicate that the introduction of palmitic acid increased the hydrophobicity of the conjugated tracer, but due to the high hydrophilicity of 3PRGD_2_, the resulting radiopharmaceutical still had an appropriate n-octanol/PBS distribution coefficient.

### 3.3. Blood Clearance Study

Blood clearance studies were performed in normal KM mice. The blood clearance curves and half-lives of ^177^Lu-Palm-3PRGD_2_, ^177^Lu-3PRGD_2_ and ^177^Lu-Palm are shown in Figure 2A and Table 2. ^177^Lu-3PRGD_2_ showed a particularly fast blood clearance (T_1/2α_ = 1.94 min; T_1/2β_ = 11.81 min), ^177^Lu-Palm showed prolonged blood retention (T_1/2α_ = 3.33 min; T_1/2β_ = 29.82 min), and ^177^Lu-Palm-3PRGD_2_ had the longest blood half-life (T_1/2α_ = 4.49 min; T_1/2β_ = 73.42 min). The result of area under the curve (AUC) analysis further suggested that ^177^Lu-Palm-3PRGD_2_ (AUC = 4002.00) had significantly improved blood retention capacity compared with ^177^Lu-3PRGD_2_ (AUC = 430.30) and ^177^Lu-Palm (AUC = 1611.00). Blood clearance studies were also performed in C57BL/6 mice. The blood clearance curves and half-lives of ^177^Lu-Palm-3PRGD_2_ and ^177^Lu-3PRGD_2_ are shown in the Appendix A. ^177^Lu-Palm-3PRGD_2_ also demonstrated significantly longer blood half-life (T_1/2α_ = 2.08 min; T_1/2β_ = 63.71 min) than that of ^177^Lu-3PRGD_2_ (T_1/2α_ = 1.06 min; T_1/2β_ = 14.54 min).

### 3.4. Competition Assays

To determine whether palmitic acid conjugation compromised the binding affinity of 3PRGD_2_ for integrin α_v_β_3_/α_v_β_5_, competition binding assays were performed (Figure 2B). The IC_50_ values were calculated to be 0.66 ± 1.20 nM for ^177^Lu-3PRGD_2_ and 5.13 ± 1.16 nM for ^177^Lu-Palm-3PRGD_2_. Meanwhile, the IC_50_ value for ^177^Lu-Palm-3PRGD_2_ was 5.82 ± 1.29 nM in the presence of HSA. The binding affinity of 3PRGD_2_ for integrin α_v_β_3_/α_v_β_5_ was affected to some extent but was still at the nanomolar level. Notably, the binding affinity of ^177^Lu-Palm-3PRGD_2_ was not different in the presence or absence of HSA, possibly due to the higher affinity of 3PRGD_2_ for integrin α_v_β_3_/α_v_β_5_ than that of palmitic acid for HSA.

### 3.5. Cell Uptake Study

The cellular uptake results for ^177^Lu-Palm-3PRGD_2_ and ^177^Lu-3PRGD_2_ are shown in Figure 2C. The uptake of ^177^Lu-3PRGD_2_ in U87MG tumor cells was 6.74 ± 0.39%, 6.88 ± 0.76% and 7.74 ± 1.24% after incubation for 1, 4 and 24 h, respectively. The uptake of ^177^Lu-Palm-3PRGD_2_ was much higher than that of ^177^Lu-3PRGD_2_, with values of 15.84 ± 1.00%, 21.30 ± 0.72% and 16.62 ± 1.08% after incubation for 1, 4 and 24 h, respectively. These results indicate that the introduction of palmitic acid might improve the cellular uptake of ^177^Lu-3PRGD_2_.

### 3.6. Small Animal SPECT/CT Imaging

The in vivo properties of ^177^Lu-Palm-3PRGD_2_, ^177^Lu-3PRGD_2_ and ^177^Lu-Palm were evaluated using small animal SPECT/CT in MC38 tumor-bearing mice. Representative images of ^177^Lu-3PRGD_2_ are shown in Figure 3A. ^177^Lu-3PRGD_2_ was rapidly cleared from circulation and excreted via the renal urinary system. Its tumor uptake was relatively low, and under the same conditions used to visualize tumor uptake of ^177^Lu-Palm-3PRGD_2_, the tumors could not be clearly visualized. The mice injected with ^177^Lu-Palm-3PRGD_2_ were imaged until 72 h p.i., and representative images are presented in Figure 3B. Tumors were clearly visible, and ^177^Lu-Palm-3PRGD_2_ maintained high accumulation and long retention in tumors from 1 to 72 h p.i. The blood pool uptake of ^177^Lu-Palm-3PRGD_2_ was obvious from 1 to 4 h p.i., showing long circulation in the blood. A blocking study was performed by co-injecting ^177^Lu-Palm-3PRGD_2_ with excess unlabeled 3PRGD_2_. SPECT/CT images and quantification of tumor uptake are shown in Figure 3C. The tumor uptake was significantly decreased (15.00 ± 0.13 %IA/cc vs. 3.22 ± 0.13 %IA/cc, *p* < 0.0001), indicating the active tumor targeting by 3PRGD_2_ (Figure 3D). Interestingly, ^177^Lu-Palm was barely detectable in the tumors (Figure 3E) and was rapidly distributed from the blood to organs (mainly to the liver) and metabolized, finally accumulating in the gall bladder before being excreted in feces.

### 3.7. In Vivo Biodistribution Study

The in vivo biodistribution properties of ^177^Lu-Palm-3PRGD_2_, ^177^Lu-Palm, and ^177^Lu-3PRGD_2_ were further evaluated in MC38 tumor-bearing mice. The tumor uptake of ^177^Lu-Palm-3PRGD_2_ increased from 14.41 ± 2.53 to 26.27 ± 6.34% IA/g at 1 h and 4 h p.i. and then decreased gradually over time with the values of 22.91 ± 4.20, 17.22 ± 3.30, and 5.83 ± 1.27 %IA/g at 12, 24 and 72 h p.i. (Figure 4A and Appendix A). ^177^Lu-Palm-3PRGD_2_ was distributed into the main organs within 4 h, resulting in reduced blood pool uptake (9.73 ± 1.16 to 4.11 ± 1.52 %IA/g at 1 h and 4 h, *p* = 0.0071). The liver and kidney demonstrated relatively high uptake, with values of 22.40 ± 2.55 and 20.11 ± 0.60% IA/g at 4 h p.i., which reduced to 6.73 ± 1.18 and 10.58 ± 1.11% IA/g at 72 p.i. Similarly, intestinal uptake was also high, with values of 20.65 ± 1.97 to 12.80 ± 1.06% IA/g at 1 h and 72 h p.i. Co-injection of ^177^Lu-Palm-3PRGD_2_ with excess 3PRGD_2_ significantly reduced tumor uptake (14.41 ± 2.53 vs. 7.00 ± 2.47% IA/g, *p* = 0.011) in the blocking study, indicating receptor-mediated uptake (Figure 4B and Appendix A). In addition, the intestinal uptake of ^177^Lu-Palm-3PRGD_2_ was notably reduced in the blocking study (20.65 ± 1.97 vs. 7.53 ± 2.15% IA/g, *p* < 0.01). Otherwise, the blood uptake was significantly higher in the blocking group (27.88 ± 4.66 vs. 9.73 ± 1.16% IA/g, *p* < 0.01) due to reduced distribution from blood to organs (such as the liver, spleen, intestine, and kidney).

^177^Lu-Palm and ^177^Lu-3PRGD_2_ were evaluated in MC38 tumor-bearing mice at 4 and 12 h p.i. (Figure 4C,D). Compared to ^177^Lu-Palm-3PRGD_2_, ^177^Lu-3PRGD_2_ exhibited relatively lower tumor uptake at 4 and 12 h p.i. (4 h: 6.22 ± 0.62 vs. 26.27 ± 6.34% IA/g, *p* = 0.0013; 12 h: 4.11 ± 0.70 vs. 22.91 ± 4.20% IA/g, *p* = 0.0003). Due to modification with palmitic acid, ^177^Lu-Palm-3PRGD_2_ showed increased retention in the blood compared with ^177^Lu-3PRGD_2_ at 4 h p.i. (4.11 ± 1.52 vs. 0.42 ± 0.26% IA/g, *p* < 0.01), resulting in relatively higher background uptake (Appendix A). Without the active targeting capability, tumor uptake of ^177^Lu-Palm was negligible at 4 and 12 h p.i. (0.80 ± 0.09 and 0.68 ± 0.05% IA/g, respectively). ^177^Lu-Palm mainly accumulated in the liver (11.00 ± 0.90 to 7.40 ± 1.15% IA/g at 4 and 12 h p.i.), was metabolized through the gall bladder and was excreted in feces (Figure 3E and Figure 4C,D).

### 3.8. Therapeutic Efficacy of Targeted Radionuclide Therapy

The therapeutic efficacies of ^177^Lu-Palm-3PRGD_2_, ^177^Lu-3PRGD_2_ and ^177^Lu-Palm were investigated in immunocompetent mice bearing MC38 tumors. The tumor uptake of ^177^Lu-Palm-3PRGD_2_ was significantly higher than that of ^177^Lu-3PRGD_2_ or ^177^Lu-Palm at all determined time points (Figure 5A). The uptake of ^177^Lu-Palm-3PRGD_2_ in tumors and major organs was calculated as AUC_0h__→72h_ (Figure 5B). The AUC values of ^177^Lu-Palm-3PRGD_2_ in MC38 tumor and the blood, kidney and liver were 1031.0 ± 97.25, 104.5 ± 7.98, 1087.0 ± 62.73 and 915.6 ± 74.74% IA/g·h, respectively, suggesting high radioactivity accumulation and retention in tumors for targeted radionuclide therapy. The results of targeted radionuclide therapy are illustrated in Figure 5C. Compared to the saline or ^177^Lu-Palm group, the ^177^Lu-3PRGD_2_ and ^177^Lu-Palm-3PRGD_2_ groups revealed significant tumor inhibition effects. Compared with ^177^Lu-3PRGD_2_, ^177^Lu-Palm-3PRGD_2_ suppressed tumor growth more efficiently due to higher tumor uptake and longer tumor retention. Transient body weight loss was observed in the ^177^Lu-Palm-3PRGD_2_ group during the initial treatment but then returned to a healthy level at the end of the treatment studies (Figure 5D). No significant body weight loss (>20% loss of the original weight) was observed in any treatment group. H & E staining of the main organs showed negligible toxic effects in all treatment groups (Appendix A).

## 4. Discussion

Integrin α_v_β_3_/α_v_β_5_ is specifically overexpressed in tumor neovascularization and a variety of tumor cells, making it an attractive target for the development of broad-spectrum targeted radiopharmaceuticals. Previously, ^177^Lu-3PRGD_2_, which targets integrin α_v_β_3_/α_v_β_5_, was developed for treatment of integrin α_v_β_3_/α_v_β_5_ positive tumors. However, the short blood half-life of ^177^Lu-3PRGD_2_ resulted in low availability in target organs. This limitation required treatment with higher or more frequent doses (111 MBq/3 mCi × 2), both of which may increase the likelihood of adverse side effects [31]. To improve the blood half-life of drugs, drug molecules can be conjugated to albumin-binding molecules to provide an extended half-life in blood [3,10,20]. Fatty acid modified peptide drugs have been widely used in clinical practice as long-acting drugs, which proves that this strategy has great potential for clinical translation [21]. At present, this strategy has not yet been used in the development of RGD radiopharmaceuticals. It is not known whether fatty acid-modified RGD radiopharmaceuticals can be sufficiently effective as long-acting radiopharmaceuticals to achieve enhanced efficacy and attenuated toxicity so that a single low-dose administration can cure tumors, guaranteeing the potential for further clinical translation. Here, we introduced the albumin binder palmitic acid into the ^177^Lu-3PRGD_2_ peptide structure to obtain the long-acting radiopharmaceutical ^177^Lu-Palm-3PRGD_2_ and evaluated its therapeutic efficacy in a tumor model.

First, palmitic acid modification inevitably affected the binding affinity or selectivity of 3PRGD_2_ to integrin α_v_β_3_/α_v_β_5_, but the IC_50_ value of ^177^Lu-palm-3PRGD_2_ was still within the nanomolar range. In addition, the cellular uptake of ^177^Lu-Palm-3PRGD_2_ was much higher than that of ^177^Lu-3PRGD_2_ in U87MG cells, suggesting that the introduction of palmitic acid might have additional effects on cell uptake. Most likely, hydrophobic interactions between the palmitic acid carbon chain and membrane lipids or lipid rafts potentially distort the outer phospholipid monolayer and accordingly induce palmitic acid internalization and enhance the cellular uptake of tracer [33,34,35]. The increased cell uptake may also contribute to increased tracer retention in tumors. Western blotting results in Appendix A showed that U87MG cells and tumor tissues were both positive for integrin α_v_β_3_/α_v_β_5_ expression, while MC38 cells and tumor tissues both had low expression of α_v_β_3_ and no integrin α_v_β_5_ expression, which was consistent with the results of previous studies [31,32,36,37]. We therefore performed in vitro cell competition binding studies using U87MG tumor cells. However, in order to better compare the enhancement of tumor efficacy caused by prolonged drug retention, MC38 cells with relatively low expression of integrin α_v_β_3_ were selected to establish tumor model and evaluate the enhancement of efficacy in vivo. If there is enough curative effect in such models, the curative effect of models with high expression of integrin α_v_β_3_/α_v_β_5_ will be more guaranteed. Moreover, MC38 is a mouse-derived cell line, and the tumor immune microenvironment of MC38 tumor bearing mice is active, which can simulate the therapeutic effect of the patient with immune response in immune-competent mice [38].

Furthermore, the blood half-life of ^177^Lu-palm-3PRGD_2_ was significantly longer than that of ^177^Lu-3PRGD_2_ (the AUC was almost 10 times higher), indicating that palmitic acid modification resulted in a significant prolongation of blood circulation time. This may be caused by several factors: first, palmitic acid can reversibly bind to albumin to prolong the blood retention time and reduce the elimination of peptides [21]; second, ^177^Lu-palm-3PRGD_2_ is amphiphilic and may self-assemble to form tiny nanoparticles (possibly nanospheres or nanofibers) with larger molecular sizes, resulting in longer blood retention times and higher tumor uptake [33,39]. Research on the hydrodynamic characteristics of palmitoylated peptides is limited, and more work is needed to further confirm our hypothesis. In addition to a significantly longer blood circulation time, tumor uptake and tumor retention of ^177^Lu-Palm-3PRGD_2_ were significantly increased. Its tumor uptake was enhanced not only by the tumor targeting capacity of 3PRGD_2_ but also by the albumin carrier characteristics of palmitic acid. The “piggy-back” strategy via reversible albumin binding of palmitic acid prolonged the ligand–receptor binding window and gave the radiopharmaceuticals more opportunities to bind to target receptors. The tumor uptake and retention of ^177^Lu-Palm-3PRGD_2_ were significantly increased compared with those of ^177^Lu-3PRGD_2_ and ^177^Lu-Palm, indicating that the effective dose and duration of action of ^177^Lu-Palm-3PRGD_2_ in tumors were greatly increased, and thus, the therapeutic efficacy of ^177^Lu-Palm-3PRGD_2_ was significantly enhanced. Total elimination of the tumors was achieved with a single injection of 18.5 MBq/0.5 mCi ^177^Lu-Palm-3PRGD_2_. This is significantly better than the efficacy achieved by two injections of 111 MBq/3 mCi ^177^Lu-3PRGD_2_ in the U87MG tumor model in our previous study [31]. Moreover, the human effective absorbed dose of ^177^Lu-Palm-3PRGD_2_ (4.04 × 10^−2^ mSv/MBq) was estimated from biodistribution data in mice by Olinda software and shown as Appendix A. The effective absorbed dose of ^177^Lu-3PRGD_2_ previously studied is 1.35 × 10^−2^ mSv/MBq [31]. Since the injected dose of ^177^Lu-Palm-3PRGD_2_ is reduced by at least 6 times (18.5 MBq/0.5 mCi vs. 111 MBq/3 mCi × 2) under a comparable therapeutic effect, its effective absorbed dose in human will be greatly reduced, and its safety will be more guaranteed than ^177^Lu-3PRGD_2_.

Notably, increased tumor uptake was accompanied by increased uptake in normal organs, which may lead to tissue damage. This is partially due to increased blood accumulation and circulation time of ^177^Lu-Palm-3PRGD_2_, resulting in higher uptake of blood rich organs such as liver and spleen. In addition, palmitic acid itself has high lipophilicity, which will also increase the uptake of metabolic organs such as liver and intestine. The biodistribution results of ^177^Lu-Palm confirmed the corresponding high liver and intestinal uptake caused by high lipophilicity. The results of the blocking study indicated that uptake was significantly inhibited in the intestine, liver, and kidney, suggesting that uptake in these tissues may be mediated in part by integrin α_v_β_3_/α_v_β_5_. In addition, two other regions showed relatively strong radioactivity accumulation at late time points (12, 24, and 72 h p.i.), probably the adrenal glands based on the shape and location of regions of interest, but the exact uptake mechanism remains unclear and requires further verification. This may be related to the introduction of albumin binding molecules, as this phenomenon was not detected in the imaging results of ^177^Lu-3PRGD_2_. Although no significant bodyweight loss and no observable tissue damage were found, more confirmation should be performed before further clinical application.

Although encouraging results have been obtained, our study also has some limitations. First, we performed in vitro studies using U87MG cells for binding affinity and specificity evaluation, but we assessed the in vivo characteristics of the probe in C57BL/6 mice bearing MC38 tumor models. Although MC38 tumor models were proved to be suitable for targeted radionuclide therapy [38], in vivo evaluation should also be completed in U87MG tumor model to verify the properties of the probe more convincingly. Second, we studied the blood clearance characteristics of RGD radiopharmaceuticals in KM normal mice, but the biological distribution characteristics of RGD radiopharmaceuticals were determined in C57BL/6 tumor bearing mice. Although these two experiments are independent, the inconsistency of mouse strains will affect the correlation and comparison of data [40]. So we then performed the blood clearance study in C57BL/6 mice and Appendix A. Even though there were some differences between strains, ^177^Lu-Palm-3PRGD_2_ showed significantly longer blood circulation time than ^177^Lu-3PRGD_2_ (4–6 times) in both mouse strains. These results confirm that the introduction of palmitic acid has led to a significant increase in the amount and time of blood retention of RGD drugs, but further pharmacokinetic studies are still needed before clinical translation research in the future. Third, we only used the MC38 tumor model in the treatment study, and the efficacy of treatment needs to be verified in more tumor models. In addition, this study only evaluated one therapeutic dose (18.5 MBq/0.5 mCi), and whether similar therapeutic effects can be achieved at ~9 MBq/250 µCi or even lower doses also needs to be further investigated in future work. Furthermore, our present study mainly focused on the development of a novel albumin binder-modified radiopharmaceutical, and the mechanism underlying its tumor growth inhibition remains to be further explored.

## 5. Conclusions

In this study, we designed and synthesized a novel long-acting integrin α_v_β_3_/α_v_β_5_-targeted radiopharmaceutical for targeted radiotherapy. Conjugation of palmitic acid to 3PRGD_2_ markedly extended its blood circulation time and enhanced tumor uptake and retention. The resulting drug conjugate showed remarkable tumor treatment efficacy without observable tissue damage. In conclusion, the introduction of palmitic acid as an albumin binder can be a promising strategy to promote small molecule peptide-based radiopharmaceuticals for targeted radiotherapy of cancer.

## Figures and Tables

**Figure 1 pharmaceutics-14-01327-f001:**
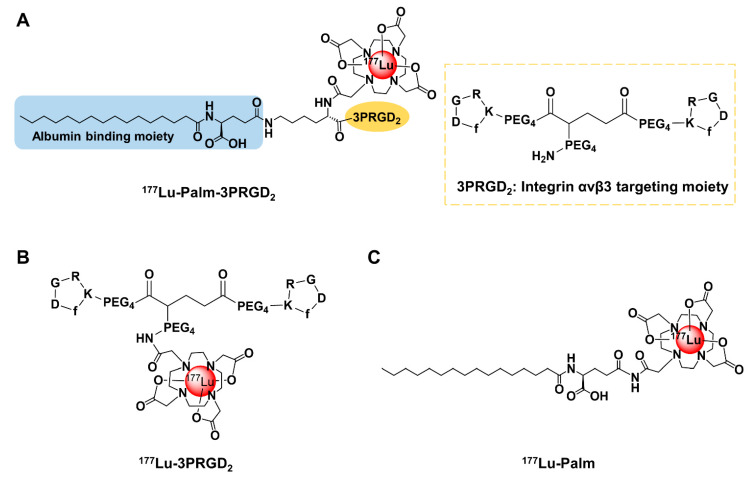
Schematic structures of (**A**) ^177^Lu-Palm-3PRGD_2_, (**B**) ^177^Lu-3PRGD_2_, and (**C**) ^177^Lu-Palm. Palmitic acid, as an albumin-binding moiety, is marked with a blue shadow. Integrin α_v_β_3_/α_v_β_5_-targeting 3PRGD_2_ is marked with a yellow shadow.

**Figure 2 pharmaceutics-14-01327-f002:**
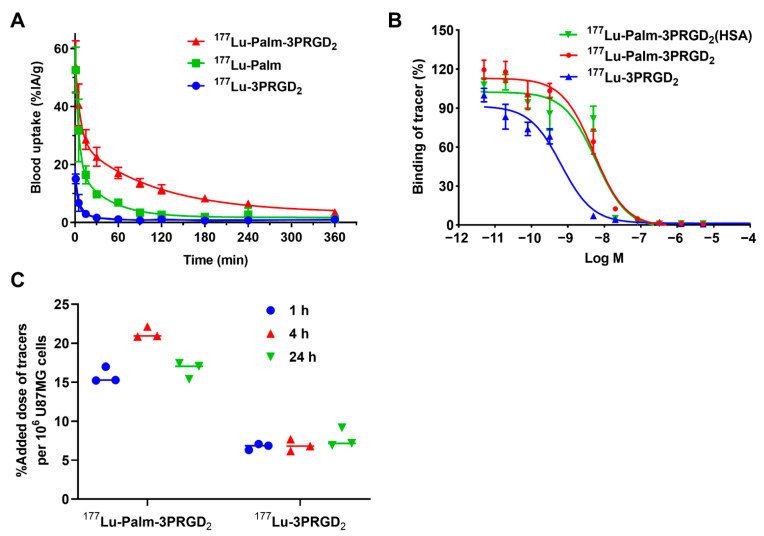
In vitro and in vivo characteristics of radiopharmaceuticals. (**A**) Blood clearance curves of ^177^Lu-Palm-3PRGD_2_, ^177^Lu-3PRGD_2_, and ^177^Lu-Palm performed in KM mice. (**B**) Competition binding assays of ^177^Lu-Palm-3PRGD_2_ (in the presence or absence of HSA) and ^177^Lu-3PRGD_2_ at different concentrations of unlabeled 3PRGD_2_ peptide. (**C**) Cell uptake of ^177^Lu-Palm-3PRGD_2_ and ^177^Lu-3PRGD_2_ at 1, 4 or 24 h after incubation with integrin α_v_β_3_/α_v_β_5_-positive U87MG glioma cells.

**Figure 3 pharmaceutics-14-01327-f003:**
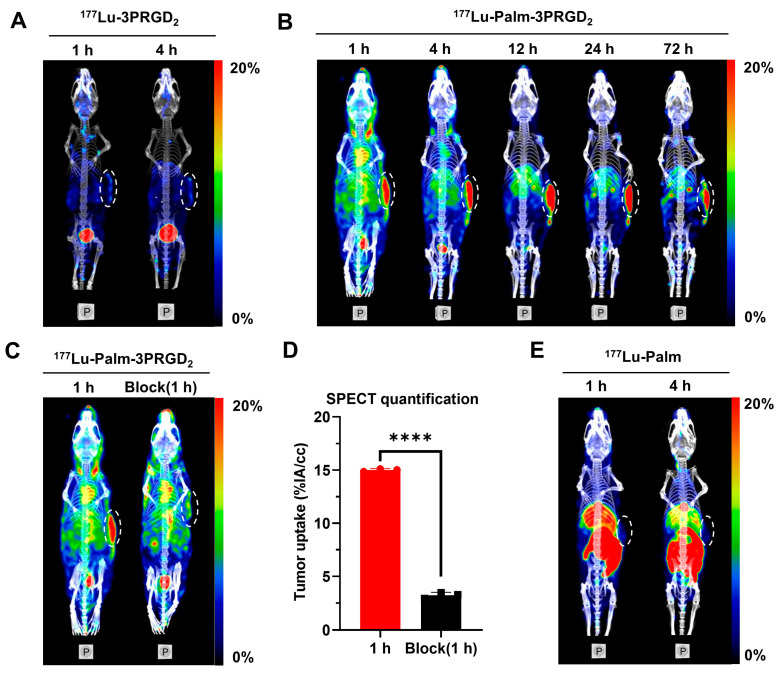
Representative nanoScan SPECT/CT images of radiopharmaceuticals in MC38 tumor-bearing mice. The images are posterior views obtained by maximum intensity projection (MIP) and fused with reconstructed CT. (**A**) Images of ^177^Lu-3PRGD_2_ at 1 and 4 h p.i. (**B**) Images of ^177^Lu-Palm-3PRGD_2_ at 1–72 h p.i. (**C**) Images of ^177^Lu-Palm-3PRGD_2_ at 1 h p.i. without or with coinjection of excess unlabeled 3PRGD_2_ as a competitor. (**D**) SPECT quantification of tumor uptake of ^177^Lu-Palm-3PRGD_2_. (**E**) Images of ^177^Lu-Palm at 1 and 4 h p.i. All tumors are circled with white dotted lines. The significance indicator **** corresponds to *p* < 0.0001 determined by Student’s *t*-test.

**Figure 4 pharmaceutics-14-01327-f004:**
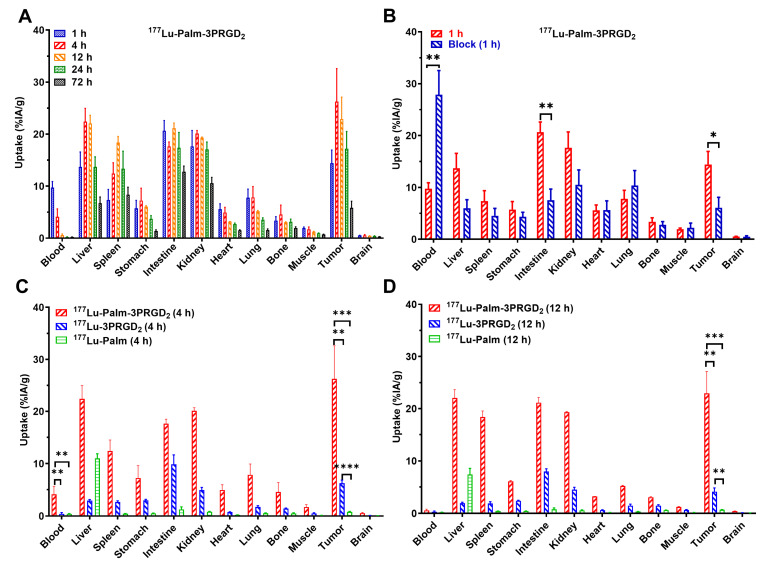
Biodistribution of radiopharmaceuticals in MC38 tumor-bearing mice. (**A**) Biodistribution results for ^177^Lu-Palm-3PRGD_2_ at 1, 4, 12, 24, and 72 h p.i. (**B**) Blocking study of ^177^Lu-Palm-3PRGD_2_ after coinjection of excess unlabeled 3PRGD_2_ peptide at 1 h p.i. (**C**,**D**) Comparison of the biodistribution results for ^177^Lu-Palm-3PRGD_2_, ^177^Lu-Palm and ^177^Lu-3PRGD_2_ at 4 h and 12 p.i., respectively. * Indicates *p* < 0.05, ** indicates *p* < 0.01, *** indicates *p* < 0.005, and **** indicates *p* < 0.0001, determined by multiple unpaired *t*-test.

**Figure 5 pharmaceutics-14-01327-f005:**
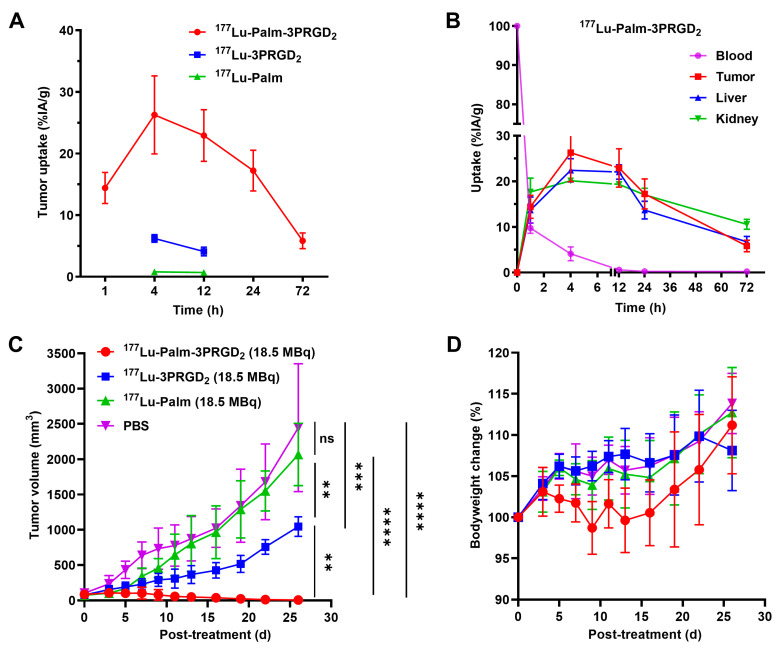
(**A**) Time-activity curves for ^177^Lu-Palm-3PRGD_2_, ^177^Lu-3PRGD_2_, and ^177^Lu-Palm in tumors. The values are based on a biodistribution study. (**B**) Time-activity curves for ^177^Lu-Palm-3PRGD_2_ in tumors and major organs. The values are based on a biodistribution study. Tumor growth curves (**C**) and body weight change curves (**D**) after targeted radionuclide therapy with PBS (as a control), ^177^Lu-Palm-3PRGD_2_ (18.5 MBq), ^177^Lu-3PRGD_2_ (18.5 MBq) or ^177^Lu-Palm (18.5 MBq) in C57BL/6 mice bearing established MC38 tumors. ns indicates non-significance, ** indicates *p* < 0.01, *** indicates *p* < 0.001, and **** indicates *p* < 0.0001 determined by Student’s *t*-test.

**Table 1 pharmaceutics-14-01327-t001:** LogP_O/W_ values and HPLC retention time of ^177^Lu-Palm-3PRGD_2_, ^177^Lu-3PRGD_2_ and ^177^Lu-Palm.

	^177^Lu-Palm-3PRGD_2_	^177^Lu-3PRGD_2_	^177^Lu-Palm
Log P_O/W_	−1.25 ± 0.07	−4.05 ± 0.14	5.93 ± 0.01
HPLC retention time/min	22.07	5.05	25.13

**Table 2 pharmaceutics-14-01327-t002:** Blood Half-life values and AUC values of ^177^Lu-Palm-3PRGD_2_, ^177^Lu-3PRGD_2_ and ^177^Lu-Palm.

	^177^Lu-Palm-3PRGD_2_	^177^Lu-3PRGD_2_	^177^Lu-Palm
Half-Life (T_1/2α_) min	4.49	1.94	3.33
Half-Life (T_1/2β_) min	73.42	11.81	29.82
AUC (%IA/g·min)	4002.00	430.30	1611.00

## Data Availability

All data generated or analyzed during this study are included in this published article and its Appendix A.

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
