# Peer review of "Palmitic Acid-Conjugated Radiopharmaceutical for Integrin αvβ3-Targeted Radionuclide Therapy"

_pharmaceutics, 2022, doi:10.3390/pharmaceutics14071327_

Round 1

Reviewer 1 Report

Yang and colleagues aimed at validating in vitro and in vivo a modified version of an anti-αvβ3 peptide, named 177Lu-3PRGD2, for target-specific cancer theragnosis. The peptide had reported by the same group as promising but showed a short plasma half-life, insufficient tumor uptake and retention, limiting its application. To circumvent these problems, in the present work, the authors proposed a modified version of the radiopeptide by introducing a fatty acid, specifically palmitic acid (Palm), and characterised the in vitro and in vivo properties of 177Lu-Palm-3PRGD2 in comparison with the unmodified 177Lu-3PRGD2.

Strengths:

  1. Preliminary data supporting the intention of using the 177Lu-3PRGD2 radiopeptide.
  2. The rationale of conjugating the radiopeptide with palmitic acid to improve its pharmacokinetics and pharmacodynamics is well exposed and convincing
  3. All the synthetic processes and analytical characterization are well reported

Weaknesses:

  1. The authors reported in the Introduction only the strengths of targeting the αvβ3 integrin, addressing the specificity of its expression by tumors and tumor blood vessels. They did not mention the problematics related to homology between other integrins such as αvβ5, nor the issues also faced in the clinic regarding the expression of αvβ3 by blood vessels of inflamed tissues, or healthy tissues, as well as by immune cells (The literature is wide. I.e., Waldeck J, Häger F, Höltke C, Lanckohr C, von Wallbrunn A, Torsello G, Heindel W, Theilmeier G, Schäfers M, Bremer C. Fluorescence reflectance imaging of macrophage-rich atherosclerotic plaques using an alphavbeta3 integrin-targeted fluorochrome. J Nucl Med. 2008 Nov;49(11):1845-51. Du F, Garg AV, Kosar K, Majumder S, Kugler DG, Mir GH, Maggio M, Henkel M, Lacy-Hulbert A, McGeachy MJ. Inflammatory Th17 Cells Express Integrin αvβ3 for Pathogenic Function. Cell Rep. 2016 Aug 2;16(5):1339-1351. doi: 10.1016/j.celrep.2016.06.065. Chinot OL. Cilengitide in glioblastoma: when did it fail? Lancet Oncol. 2014 Sep;15(10):1044-5. doi: 10.1016/S1470-2045(14)70403-6. Jenkins WS, Vesey AT, Stirrat C, et al. Cardiac αVβ3 integrin expression following acute myocardial infarction in humans. Heart. 2017;103(8):607-615. Alday-Parejo B, Stupp R, Rüegg C. Are Integrins Still Practicable Targets for Anti-Cancer Therapy?. Cancers (Basel). 2019;11(7):978). These problematics are recurrent in the clinic and limited the use of αvβ3-specific radio/pharmaceuticals, and must be taken into consideration in the preclinical validation of new αvβ3 targeting molecules.

In light of these problematics, overall, the in vitro and in vivo studies are not well designed and performed. Just for giving some examples:

  • In vitro and in vivo experiments were performed only on one cell line, respectively: U87MG in vitro, and MC38 in vivo. Since references nor expression analysis were reported, they assumed these cells express αvβ3. Nevertheless, the authors did not performed studies attesting the specificity of the radio peptide by adding a αvβ3-negative and a αvβ3-negative but αvβ3-positive cell lines. Therefore, receptor specificity was not well investigated, and I would like to highlight how it is quite unprofessional to base the in vivo validation of a new drug entity only on one receptor-positive tumor-bearing mouse model.
  • Biodistribution studies in healthy mice are completely lacking.
  • Since αvβ3 is expressed by different immune cells, toxicity assessment should also include lymphocyte counts.
  • The specificity of targeting αvβ3 may be very low due also to upregulation of αvβ3 integrin on inflamed blood vessels. Chemotherapy or pre-existing inflammation, especially in aging populations, can generate significant off-targeting events. Control groups in the animal studies could be designed to account for this.
  • Statistical analysis should be checked. In Figure 4 there are data that appear to be statistically significant but are not reported to be.

  1. References must be checked. For example, the authors reported: “Our 99mTc-labeled 3PRGD2 has been clinically used as a diagnostic tracer for early detection of various tumors, and its clinical phase III study has been completed [28–30]” (lines 59-60). However, the reference 28 referred to animal studies.
  2. English must be revised. Sometimes it is quite colloquial, and vocabulary is repetitive (i.e., describing results the verb “show” was quite abused). Please use synonyms.

Reviewer 2 Report

This is a well structured and written study 

Reviewer 3 Report

The article by Yang et al. describes the development and preclinical evaluation of a 177Lu-labeled RGD dimer conjugate for potential therapy of integrin alpha3betav expressing tumors and tumor vasculature. The authors modified their previously described RGD dimer by an albumin binder, in this case palmitic acid, to prolong blood circulation and, thus, tumor uptake and retention. The study encompasses the chemical syntheses of the palmitic acid dimer with a DOTA chelator as well as a control lacking the RGD moieties and the previously described compound without the palmitic acid, labeling with 177Lu, log D determination, competitive binding measurements and cellular uptake studies in U87MG cells, metabolic studies as well as ex vivo biodistribution and SPECT imaging studies in mice. Finally, the therapeutic efficacy of the novel bioconjugate has been evaluated in xenograft mice showing tumor growth inhibition after a single treatment.

This is a sound study, all relevant experiments that are recommended for the preclinical evaluation of a novel radiotracer have been performed. Their description is (mostly) detailed enough to allow their reproduction. The manuscript is well written and appropriate in length. The topic is certainly of interest for the readership of Pharmaceutics. The fact that numerous RGD-based radiotracers with excellent pharmacokinetics have been described in the past, lowers the novelty of the manuscript to some extent but the results of this study are still relevant to the nuclear medicine and radiopharmaceutical community.

Nonetheless, this reviewer has some questions and comments that should be addressed before acceptance for publication.

Major

  • As pointed out above, the study is well performed. However, the main point of criticism is that the introduction of the albumin-binding moiety does not really improve the performance of the radiotracer. The authors themselves discuss the limitation briefly but the question remains “Is this radiotracer really superior to the previously described RGD dimer without albumin binding site?”. When looking at the tumor-to-tissue ratios, the prolonged blood circulation led to increased tumor uptake but also to higher uptake in non-target tissues such as the liver, spleen, intestine, bone, etc., resulting in inferior T/B ratios. Despite the authors didn’t notice any severe side effects or organ damage after a single treatment in this report, it remains questionable if the novel compound is really an improvement. For example, when looking at the spleen, a radiation sensitive organ, the tumor-to-spleen ratio is 1.33 for the albumin-binding conjugate at 12 h p.i., whereas it is 2.17 for the previously described dimer. An identical finding can be seen for the liver with 1.04 vs. 2.13, respectively. Similarly, the ratios already worsened at earlier time points (4 h p.i.). Very problematic might also be the bone uptake when aiming for PRRT application. When comparing the bone uptake at 1 h p.i. with the blocking group (3.39 % vs. 2.78 %), it becomes obvious that there is a significant amount of unspecific accumulation in the bone (and all other organs), which is problematic for future applications in patients. Even at 72 h p.i. about 3% of activity remains in the bone, which corresponds to 180 MBq Lu-177 assuming a “standard” therapy dose of 6 GBq. Moreover, the uptake in the bone is clearly visible in the SPECT images showing the shoulder joint, even at 72 h p.i. Furthermore, the prolonged retention in tumors is also seen in healthy tissues demonstrating that the observed uptake is mostly unspecific.

Could the authors please comment on the reviewer’s concerns?

The authors are also asked to include the tumor-to-tissue ratios into the corresponding tables such that the readership has this information directly at hand.

  • In figure 2, the cell uptake for the novel and the previous RGD dimer are given. It would have been important to include a blocking study in these experiments to determine the amount of unspecific cell binding. Also, please include how the percentage of added radioactivity was measured? Were standards used? If so please include details.
  • The introduction of the palmitic acid resulted in a ~10fold decrease of receptor affinity, this is certainly not a slight affections as stated in line 399.

Minor

  • The authors are asked to use the “consensus nomenclature rules for radiopharmaceutical chemistry” Nucl Med Biol 2017;55:v-xi.doi: 10.1016/j.nucmedbio.2017.09.004. In particular in naming their labeled compounds as well as the term “specific” activity, which should be “molar” activity according to the standardized rules.
  • 3. radiochemistry and in vivo stability: how much activity was administered per mouse? Please also provide the molar amount of conjugate.
  • 7. cell uptake assay: How many cells were seeded per well? Please also provide, the activity and molar amount added to each well.
  • 8., line 196: please rephrase the expression “in the absence of increasing concentrations of ….”, not clear what this means, decreasing amounts of? Also, see comment for 2.7.
  • Please provide molar amounts of peptide injected for 2.9, 2.10 and 2.11.
  • Please check the y-axis of figure 2C, added instead of adding?

Reviewer 4 Report

In the present manuscript, the research group studies the uptake and therapeutic effect of a modified αvβ3 ligand. For this purpose, palmitic acid was coupled as an albumin-binding group to the promising peptide 3PRGD2. Subsequently, the binding, distribution and accumulation of the ligand and corresponding controls were analyzed in vitro and in vivo, respectively, and the therapeutic effect was investigated.

General:

The manuscript is well developed in terms of language and methodology. While the individual techniques for characterization are well chosen, only the consistency of the models is somewhat limited. For example, the in vitro studies were performed on human glioma cells (U87MG), whereas a mouse tumor cell line (MC38) was used as a model in the animal study.

In addition, blood stability and retention were analyzed in KM mice, but biodistribution was analyzed in C57BL/6 mice.

The authors already recognize the limitation that the in vivo studies were only performed with the MC38 model. It would be beneficial if at least some of the in vitro experiments were also done for the MC38 cells to get better consistency.

Can the blood data from the C57BL/6 biodistribution analysis of 177Lu-Palm-3PRGD2 be added to the results of blood clearance in KM mice? (3.3 and Figure 2A)

The authors should consider speaking in general of activity rather than dose (%IA/g or %IA/cc) rather than (%ID...), since in most cases activity rather than dose is spoken of in terms of the correct definition. Although this variant (%ID…) is common in the relevant literature, the use of the more correct variant should be preferred.

Cf. https://www.epa.gov/radiation/radiation-terms-and-units

Methods:

2.6

Please include the ATCC ordering numbers for the respective cell types.

Figure 3B

Based on the biodistribution data, I would expect a more prominent signal from the kidneys in the MIP at least after 1h and after 72h? Please discuss this discrepancy!

Already after 12h, but especially after 24h and 72h 2 small regions with strong accumulation are visible in the MIP - do you have an explanation of what these areas might be? (adrenal glands?)

Figure 3E:

The scale in this figure runs to 200%, while in all other figures the upper limit is set to 20%. Please check whether it would be reasonable and possible to aim for a standardization of the representation here as well, in order to make clear that no unspecific accumulation has occurred.

Discussion: page 10 Line 385-386

Sentence: "Previously, 177Lu-3PRGD2, which targets integrin avß3, was developed for treatment of U87MG tumor xenografts." Please check, if you really developed the tracer for mouse treatment, as "xenograft" suggests?

Please also discuss the strong accumulation in liver, kidney, small intestine, and spleen, which are prominent in the biodistribution. In time course, the accumulation in these organs strongly resembles the tumor signal. For example, is it specific off-target binding, non-specific binding or accumulation of metabolites?

Round 2

Reviewer 1 Report

Thanks to the authors for the reply and specification of some of my doubts. However, I do not agree with two comments.

Specifically:

"Many studies and our previous studies have verified that human glioma cells (U87MG) are integrin αvβ3 positive, and a large number of preclinical studies have used this cell for evaluation. "

I know and agree, however the cellular expression of different proteins/receptors may vary significantly between the same cell line mantained in different laboratories, as unwanted clonal selections can inadvertently take place. Indeed, many journals specifically ask for genomic validation if the cell line has been bought more than 2 years from ATCC. Therefore, it would be better to double check the expression analysis and report it in your present work, otherwise to cite your published paper in which you reported the information.

“…all both αvβ3 and αvβ5 are overexpressed on many tumor endothelial cells and tumors, and their expression is much lower in normal tissues, so the presence of αvβ5 does not affect the detection of tumors by targeting αvβ3”

“This paper focused on the effect of extending the blood circulation of 177Lu-3PRGD2 by introducing palmitic acid, an albumin-binder, to enhance its tumor uptake and treatment, so that a lower dose of administration can effectively treat tumors. Therefore, the controls used were the unmodified 177Lu-3PRGD2 and the directly labeled albumin binder, respectively. Differences between integrin αvβ3-positive and negative models were not the objective of this study, so no further studies have been done in this regard. However, we will supplement the treatment of more tumor models in future work”

I understand your position, however, I do not agree with your statement regarding αvβ5 expression, as the protein is quite expressed by many healthy organs (e.g., lung, stomach, liver, breast, spleen, ovary, as well as by blood vessels during normal angiogenesis and in inflammatory conditions). Therefore, using a cell line expressing only αvβ5 for in vivo experiments is mandatory to assess the specificity for αvβ3, especially when you are validating a pharmacophore with a prolonged blood retention, as this condition may facilitate off-target events.

Reviewer 3 Report

✅ 

Author Response

Thank you for your approval.

Reviewer 4 Report

Regarding answers of the first review:

The answers to the questions from the first revision were not sufficient or the authors did not also transfer their answers into the manuscript. Corresponding additions to most points are still missing here.

1.       The manuscript is well developed in terms of language and methodology. While the individual techniques for characterization are well chosen, only the consistency of the models is somewhat limited. For example, the in vitro studies were performed on human glioma cells (U87MG), whereas a mouse tumor cell line (MC38) was used as a model in the animal study. The authors already recognize the limitation that the in vivo studies were only performed with the MC38 model. It would be beneficial if at least some of the in vitro experiments were also done for the MC38 cells to get better consistency.

Reply: Thanks for the comments. Our previous studies have verified that human glioma cells (U87MG) are integrin αvβ3 positive, and a large number of preclinical studies have used this cell for evaluation (I.e., Improving tumor-targeting capability and pharmacokinetics of 99mTc-labeled cyclic RGD dimers with PEG4 linkers. Mol Pharm. 2009;6(1):231-45.; 2-Mercaptoacetylglycylglycyl (MAG2) as a Bifunctional Chelator for 99mTc-Labeling of Cyclic RGD Dimers: Effect of Technetium Chelate on Tumor Uptake and Pharmacokinetics. Bioconjug Chem. 2009,20(8):1559-1568.; Blood clearance kinetics, biodistribution, and radiation dosimetry of a kit-formulated integrin αvβ3-selective radiotracer 99mTc-3PRGD2 in non-human primates. Mol Imaging Biol. 2011;13(4):730-6.; Impact of bifunctional chelators on biological properties of 111In-labeled cyclic peptide RGD dimers. Amino Acids. 2011 Nov;41(5):1059-70.; Evaluation of 111In-Labeled Cyclic RGD Peptides: Effects of Peptide and Linker Multiplicity on Their Tumor Uptake, Excretion Kinetics and Metabolic Stability. Theranostics. 2011;1:322-40.; Improving tumor uptake and pharmacokinetics of 64Cu-labeled cyclic RGD peptide dimers with Gly3 and PEG4 linkers. Bioconjug Chem. 2009, 20(4):750-759.; PET Imaging of Neovascularization with 68Ga-3PRGD2 for Assessing Tumor Early Response to Endostar Antiangiogenic Therapy. Mol Pharm. 2014;11(11):3915-22.; 68Ga-Labeled 3PRGD2 for Dual PET and Cerenkov Luminescence Imaging of Orthotopic Human Glioblastoma. Bioconjug Chem. 2015;26(6):1054-60.; Two 90Y-Labeled Multimeric RGD Peptides RGD4 and 3PRGD2 for Integrin Targeted Radionuclide Therapy. Mol Pharm. 2011, 8(2):591-599.; Anti-tumor effect of integrin targeted 177Lu-3PRGD2 and combined therapy with endostar. Theranostics. 2014, 4(3):256-66.). Therefore, this cell is selected for the in vitro affinity verification of the probe.

MC38 cells themselves are cells with low expression of integrin αvβ3, so they were not used for in vitro cell experiments. However, studies have shown that MC38 tumors are integrin αvβ3-positive tumors, and integrin αvβ3 is overexpressed in tumor angiogenesis (Molecular Magnetic Resonance Imaging of Angiogenesis In Vivo Using Polyvalent Cyclic RGD-Iron Oxide Microparticle Conjugates. Theranostics 2015, 5, 515-529.), so it is more suitable for the study of RGD-targeted drug efficacy enhancement. Moreover, MC38 is a mouse-derived cell, and its tumor immune microenvironment is active (Integrin αvβ3-Targeted Radionuclide Therapy Combined with Immune Checkpoint Blockade Immunotherapy Synergistically Enhances Anti-Tumor Efficacy. Theranostics 2019, 9, 7948-7960.), which can simulate the therapeutic effect of the patient with immune response in immune-competent mice, so it is selected as the tumor model in our paper.

***2nd revision:

The reviewer does not question that the cell lines in themselves are suitable for the experiments. However, it is common practice to perform in vitro and in vivo experiments with the same cell lines to achieve some consistency. Also, performing blood stability and retention time in a different mouse strain (KM) than the biodistribution analysis (C57BL/6) is not common practice.

It would be ideal to also perform at least part of the in vitro studies in MC38 and add these results or amend the discussion in the manuscript accordingly to legitimize the use of the two different cell lines.

2.       In addition, blood stability and retention were analyzed in KM mice, but biodistribution was analyzed in C57BL/6 mice. Can the blood data from the C57BL/6 biodistribution analysis of 177Lu-Palm-3PRGD2 be added to the results of blood clearance in KM mice? (3.3 and Figure 2A)

Reply: Thanks for the comment. The blood data from the C57BL/6 model cannot be directly added to the blood clearance data from the KM mouse model. Mainly because C57BL/6 mice are tumor models, and the high tumor uptake of 177Lu-Palm-3PRGD2 has a large effect on their blood retention. In blocking studies, blood retention was significantly increased as tumor uptake was blocked. We will investigate the differences in pharmacokinetics in more models in future studies.

***2nd revision:

As mentioned earlier, the use of different mouse strains within a study is inconsistent and should at least be mentioned in the discussion. If the biodistribution data do not match the stability tests and the data of the different mouse strains do not match, this should be discussed convincingly. Here, an effect due to the tumor may be as important as the difference between the two mouse lines1. The discussion should be extended accordingly.

[1] Fuyi Xu, Tianzhu Chao, Yiyin Zhang, Shixian Hu, Yuxun Zhou, Hongyan Xu, Junhua Xiao, Kai Li, "Chromosome 1 Sequence Analysis of C57BL/6J-Chr1KM Mouse Strain", International Journal of Genomics, vol. 2017, Article ID 1712530, 9 pages, 2017. https://doi.org/10.1155/2017/1712530

3.       The authors should consider speaking in general of activity rather than dose (%IA/g or %IA/cc) rather than (%ID...), since in most cases activity rather than dose is spoken of in terms of the correct definition. Although this variant (%ID…) is common in the relevant literature, the use of the more correct variant should be preferred. Cf. https://www.epa.gov/radiation/radiation-terms-and-units

Reply: Injected dose/g (%ID/g) or injected activity/g (%IA/g) are the same because injected dose is quantified by activity. The common unit in the research field of nuclear medicine is %ID/g, so there is no need to change it.

***2nd revision:

In science, the activity is given in the unit Bq and the (absorbed) dose in the unit Gy. This can be found in very general literature such as the SI Units Guide 2008.

This reviewer does not consider it a reasonable or even admissible argument that others also use incorrect terminology.

4.       Figure 3B: Based on the biodistribution data, I would expect a more prominent signal from the kidneys in the MIP at least after 1h and after 72 h? Please discuss this discrepancy! Already after 12 h, but especially after 24 h and 72 h, 2 small regions with strong accumulation are visible in the MIP - do you have an explanation of what these areas might be? (adrenal glands?)

Reply: Albumin-modified drugs bind to albumin in the blood, thereby prolonging the blood circulation time. Due to the large size of the bound protein, it is difficult to metabolize from the kidneys, so the uptake by the kidneys is significantly reduced. This is why, although it is a metabolic organ, the uptake by the kidneys is not high. From the shape and location, we also believe that these two small regions with stronger accumulation may be the adrenal glands, and we are also interested in testing this in future studies.

***2nd revision:

Also on this point, the reviewer recommends a comment in the discussion in addition, e.g., regarding adrenals.

5.       Please also discuss the strong accumulation in liver, kidney, small intestine, and spleen, which are prominent in the biodistribution. In time course, the accumulation in these organs strongly resembles the tumor signal. For example, is it specific off-target binding, non-specific binding or accumulation of metabolites?

Reply: Thanks for the comments. In this study, palmitic acid was used as an albumin binding agent for radiotherapy drugs to increase blood accumulation and circulation time to increase the utilization of radionuclides, which also lead to higher background uptake in blood-rich organs such as liver and spleen. In addition, palmitic acid itself has high lipophilicity, which will also increase the uptake of metabolic organs such as the liver and intestine. This is also confirmed from the biodistribution results of 177Lu-Palm. Besides, the high uptake of 177Lu-3PRGD2 and 177Lu-Palm-3PRGD2 in small intestine also related to the expression of integrin αvβ3 in the small intestine. The uptake in the kidney is due to the excretion. Although uptake was increased in these normal organs, the effective dose of 177Lu-Palm-3PRGD2 in tumors increased more, and its effective absorbed dose in humans estimated from mouse biodistribution data (4.04 × 10-2 mSv/MBq) have been provided in the Supplementary Information (Table S6). The effective absorbed dose of 177Lu-3PRGD2 in humans was previously studied at 1.35 × 10-2 mSv/MBq (Anti-tumor effect of integrin targeted 177Lu-3PRGD2 and combined therapy with endostar. Theranostics. 2014, 4(3):256-66.). Since the injected dose of 177Lu-Palm-3PRGD2 is reduced by at least 6 times (18 MBq vs 111 MBq × 2) under a comparable therapeutic effect, its effective radiation absorbed dose in humans will be much lower and safer.

***2nd revision:

Once again, a corresponding modification of the discussion is missing. If the reviewer's questions can be answered, they should also appear accordingly in the discussion so that future readers of the manuscript do not have to ask these questions in the first place.

There is still a lack of consideration of the strong accumulation in the off-target organs in the discussion.

It should also be explained why the blocking studies can also have an influence on these organs. If here with the blocking the specificity of the ligand should be proven, the question arises whether the uptake in liver, intestine and kidney is also specific, because the blocking seems to have a similar effect in these organs as in the tumor! (Liver 13.70 à 5.97; Intestine 20.65 à 7.53; Kidney 17.65 à 10.48) maybe also Spleen (7.35 à 4.53)

How do you explain the increased accumulation in the lung in the blocking studies?

 6.       Please check the numbers and explain the discrepancies in the supplementary tables S1 and S2 for liver, intestine and kidney at 4h p.i..

S1

Liver

22.40

2.55

Intestine

17.63

0.87

Kidney

20.11

0.60

S2

Liver

25.48

5.64

Intestine

21.33

6.44

Kidney

24.00

6.75  

7.       Page 15, lines 437-443:

In the Discussion, the authors referred to an addition in the supplementary in which they calculated the human effective absorbed dose based on mouse biodistribution data using Olinda software. Corresponding amendments in the methods section are missing, among others, the software version used as well as the referencing of the software.

Round 3

Reviewer 1 Report

The authors did not answered to the critical issues raised, but only by agreeing in to check and perform the proposed essays the future . However, the periodic control of receptors expression by cell lines lines is essential for the creation of the basis for the biological validation of new drug entities. Moreover, in vivo esperiments assessing the selectivity towards αvβ3 compared to αvβ5 is mandatory to justify the innovation of the proposed work.

Since the authors do not want to provide these data, the present work is incomplete and therefore should be rejected.

Reviewer 4 Report

Page 3 line 171

- please correct "percentage injected dose per gram" to "percentage injected activity per gram"

Page 4 line 280

- please correct "percentage injected dose per gram" to "percentage injected activity per gram"

Page 5 line 246-249

- If the authors used Graphpad Prism for the statistical analyses mentioned, please also provide the correct citation for the software:

https://www.graphpad.com/guides/prism/latest/user-guide/citing_graphpad_prism.htm

page 12 line 429-431

revision of the sentence sugessted like: "In another study that we are currently submitting, MC38 cells were found to have low expression of integrin αvβ3 and therefore were not used for in vitro experiments in this study."

page 12 line 435

- do you mean for example "expression of integrin" instead of "expression of tumor cells"? revision of the sentence suggest

page 12 line 436

revision of the sentence is recommended, like "...MC38 is a mouse-derived cell line..." it's probably not a single cell

page 12 line 439

please correct "animal model" to "tumor model" as MC38 is not a mouse strain

page 12 line 471-473

careful revsison of the sentence is recommended, maybe like "The results of the blocking study indicated that uptake was significantly inhibited in the intestine, liver, and kidney, suggesting that uptake in these tissues may be mediated in part by integrin αvβ3."

page 12 line 474

Please formulate the sentence more carefully, because it is a plausible assumption that it is the adrenal glands, but the clear evidence is still missing!

page 13 line 490

For the future reader, there will not be a version where additional blood clearance data is not yet included in the supplementary! Revision of the sentence "We then supplemented..." is recommended!
